# How Should Transformers Encode Numeric Values in Electronic Health Records?

**Maria Elkjær Montgomery** [1]  **Christian Igel** [1]  **Mikkel Odgaard** [1]  **Martin Sillesen** [2]  **Mads Nielsen** [1]

## Abstract

How do we encode numeric values in transformer-based sequence processing, particularly in electronic health record (EHR) data? We systematically compare discrete, continuous, and hybrid value encoding strategies using synthetic arithmetic tasks embedded within real-world EHR data, as well as real-world clinical prediction tasks. Our study reveals trade-offs between numeric precision, optimisation stability, and architectural flexibility. We find that approaches that explicitly model value-concept interactions perform best on precision-sensitive arithmetic tasks when architectural constraints permit. Hybrid token-based approaches that retain numeric values but apply binning prior to projection provide a more robust and broadly applicable alternative, with the optimal number of bins following a simple empirically derived power-law in dataset size. Across tasks, models consistently exhibit reliable "good enough" numeric computation rather than exact arithmetic, while clinical gains from incorporating laboratory values are task-dependent. This suggests that robustness and deployability often outweigh maximal numeric precision in practice, motivating hybrid token-based approaches as a practical default.

## 1. Introduction

There is a growing body of research leveraging Electronic Health Record (EHR) data for predictive modelling of clinical outcomes. Early approaches primarily relied on traditional machine learning models such as XGBoost and logistic regression (Shillan et al., 2019; Stevens et al., 2023; El Haji et al., 2023; Mishra et al., 2024), but with the advent of transformer-based architectures (Vaswani et al., 2017), these methods have been extended to EHR data as well. In particular, models inspired by the BERT architecture (Devlin et al., 2019), such as BEHRT (Li et al., 2020) and Med-BERT (Rasmy et al., 2021), marked a major shift in this field, motivating subsequent work in the field (Pang et al., 2021; Steinberg et al., 2024; Wornow et al., 2025; Odgaard et al., 2024; Kraljevic et al., 2024). Despite these advances, most large-scale pre-trained models for EHR data rely primarily on structured categorical data such as diagnosis codes, procedure codes, and medication records, while continuous numeric data, such as laboratory test results or risk scores, are often underutilised. This omission is notable given that lab values are among the most routinely collected clinical measurements, typically requiring no additional documentation from clinicians and thus being less prone to subjective bias and human errors. Moreover, they constitute a rich and high-volume data source that could enhance patient representation learning.

Therefore, recent work has proposed several strategies for incorporating numeric values into transformer-based models for EHR data. These include value discretisation (Rossi et al., 2019; Renc et al., 2024; Mbaye et al., 2025), separate embedding layers for numeric features (Li et al., 2020), and joint embeddings that align categorical and continuous features (Bellamy et al., 2025). More recently, Guo et al. systematically evaluated joint embedding strategies against factorised embedding approaches for numeric encoding in clinical transformers, reflecting the growing focus on how numeric values should be encoded and integrated within transformer architectures (Guo et al., 2026). Parallel work has focused on self-supervised transformers that learn numeric encodings directly from longitudinal EHR context, with the resulting encodings then used for downstream predictive tasks (Heilbroner et al., 2025). While these approaches demonstrate potential, their reported effectiveness varies considerably, and there has been limited investigation into the conditions under which each method succeeds or fails. A further limitation is that existing studies rarely connect methodological choices to the clinical contexts in which numeric values arise. In practice, numeric features can convey very different kinds of information: a single mea-

[1]Department of Computer Science, University of Copenhagen, Copenhagen, Denmark [2]Department of Organ Surgery and Transplantation, Copenhagen University Hospital - Rigshospitalet, Copenhagen, Denmark. Correspondence to: Maria Elkjær Montgomery <maem@di.ku.dk>.

*Proceedings of the $43^{rd}$ International Conference on Machine Learning*, Seoul, South Korea. PMLR 306, 2026. Copyright 2026 by the author(s).

surement may be critical, combinations of values may carry joint significance, and temporal trends may reveal important patterns. Whether current approaches can robustly capture these distinct scenarios, especially within the finite context windows of transformers, remains unclear. Systematic evaluation across different types of numeric data distributions is therefore needed to understand both the strengths and limitations of existing methods in terms of ease of use, performance under low-data settings, and performance on metrics.

The main contributions of this study are:

- **A unified evaluation framework for numeric reasoning in EHR transformers.** We introduce a reusable test suite for systematically evaluating numeric value encodings in transformer-based EHR models. The framework combines synthetic arithmetic tasks embedded within real EHR sequences with real-world clinical prediction tasks, enabling controlled analysis of numeric reasoning capabilities under clinically relevant conditions.

- **Systematic comparison of numeric value encodings.** We present a carefully designed empirical study of discrete, continuous, and hybrid numeric value encodings in transformer-based EHR models, characterising their trade-offs in numeric precision, optimisation stability, and architectural flexibility using both synthetic arithmetic tasks embedded in real data and real-world clinical prediction.

- **Characterisation of the precision-stability trade-off.** We show that approaches that explicitly model value-concept interactions provide a strong inductive bias for fine-grained numeric reasoning, while hybrid sequence integrated approaches with binning offer a robust and scalable alternative.

- **Evidence for robust approximate numeric reasoning.** Despite prior reports that transformers struggle with exact arithmetic, we find that all evaluated methods can reliably perform approximate numeric computations, with performance degrading smoothly as task complexity increases rather than failing abruptly. This suggests that "good enough" numeric reasoning may be sufficient for EHR applications.

- **Implications for numeric reasoning in clinical prediction.** We find that clinical gains from incorporating numeric values are modest and task-dependent in long-term risk prediction, indicating that encoding choices primarily affect robustness and deployability rather than yielding large improvements in predictive performance in this setting. These effects may differ over shorter time horizons, where lab values may not have been translated into diagnostics and medication.

## 2. Related Work

### 2.1. Numeric Values in Transformers

Many studies have explored how to handle both continuous and discrete inputs within transformer architectures. As an early example, Huang et al. introduced TabTransformer (Huang et al., 2020), where categorical features are passed through attention layers to learn contextualised representations, which are then concatenated with numeric features before the prediction head. This approach achieved strong performance; however, because only categorical features participate in the attention mechanism, it captures limited interactions between numeric and categorical variables.

The FT-Transformer, proposed by Gorishniy et al. (Gorishniy et al., 2021), tokenises both categorical and continuous features into embeddings and feeds these jointly into a transformer encoder, but does not employ any pre-training. SAINT follows a similar design, representing all features as tokens, while further introducing intersample attention and a contrastive self-supervised pretraining objective (Somepalli et al., 2022). xVal, introduced by Golkar et al.(Golkar et al., 2023), extends numeric encodings by introducing an artificial token for numbers whose embedding is scaled by the numeric value, allowing the transformer to model continuous magnitudes directly.

In the context of EHR data, Labrador (Bellamy et al., 2025) adopts an approach conceptually similar to FT-Transformer and xVal: categorical and continuous features each pass through dedicated embedding layers, and their outputs are summed to form joint embeddings. This architecture was applied to laboratory test data, where each lab code and corresponding value was jointly embedded. While Labrador excelled at imputing masked lab values, it did not outperform the gradient boosting baseline XGBoost on downstream clinical prediction tasks. Additionally, as the paper itself states, the model did not include other data categories from EHR sources, and as such may not have provided a comprehensive view of the patients. Guo et al. (Guo et al., 2026) also performed a systematic comparison of joint and factorised strategies for numeric encoding in clinical transformers, reporting strong downstream performance for joint embedding approaches across a comprehensive range of clinical prediction tasks. However, their study focused exclusively on quantised numeric inputs and provided limited insight into the conditions under which different encoding strategies succeed or fail. More broadly, several EHR-focused approaches incorporate laboratory measurements by discretising continuous inputs (Rossi et al., 2019; Mbaye et al., 2025; Guo et al., 2026). These approaches often rely on domain knowledge, such as thresholds distinguishing normal from abnormal results, which may not scale to more general EHR tasks that include millions of heterogeneous numeric variables. Similarly, ETHOS, proposed by Renc

et al. (Renc et al., 2024), discretises continuous features through quantile-based binning (typically using ten quantiles) before embedding them.

Overall, a variety of strategies have been proposed for encoding numeric data, but only a subset has been systematically evaluated on EHR datasets. These schemes are also often paired with different architectures, data modalities, and evaluation protocols, limiting direct comparability. To address this, we introduce a unified test suite for comparing key approaches to encoding categorical and continuous data across a set of well-defined prediction tasks, enabling systematic evaluation of their relative performance and generalisability.

## 3. Methods

### 3.1. Data

To evaluate methods for incorporating numeric values, we use real-world EHR data and real-world sequences interjected with synthetic signals. Synthetic signals enable controlled manipulation of complexity, such as the number of features, interactions, and temporal dependencies, while the clinical data serves to validate these findings under realistic background noise and feature interdependencies. The real data are derived from the EHR system covering the Capital Region and Region Zealand in Denmark, and include all hospital interactions from 2016 to 2024 (2,218,028 unique individuals). These were approved by the Danish Patients Safety Board (Styrelsen for Patientssikkerhed, approval #31-1521-182) and the Danish Capital Region Data Safety Board (Videncenter for data-anmeldelser, approval #P-2020-180).

### 3.2. Model

In this paper, we use a BERT-style architecture inspired by CORE-BEHRT (Odgaard et al., 2024), replacing the original backbone with ModernBERT (Warner et al., 2025). CORE-BEHRT provides a strong reproducible baseline while lacking native support for numeric values, making it suitable for controlled evaluation of encoding strategies. As Figure 1 shows, patient trajectories are constructed from raw electronic health record data by extracting clinical concepts, including diagnostic codes, medication codes, procedure codes, and laboratory test names, denoted $D_x$, $M_x$, $P_x$, and $LAB_x$, respectively, where subscripts independently number the order of the unique occurrence. We also include background patient information (gender, birthdate) and visit-specific metadata such as visit time (position embedding), visit index (segment embedding), and age at each visit. All codes are tokenised and embedded together with their associated contextual information, allowing the model to process the full patient history, where individual codes correspond to words, visits to sentences, and the complete medical history to a document. The model is trained in two

stages: pre-training followed by task-specific fine-tuning. During pre-training, we employ masked language modelling (MLM), where a subset (15%) of tokens is masked and the model is trained to predict the original tokens. For fine-tuning, patient trajectories are right-censored at a predefined timestamp, and only events occurring prior to this censoring point are provided as input. A special censoring token, annotated with age and time at censoring, is appended to each sequence, and the model is trained on a binary classification task using a BiGRU-based (Zhao et al., 2017) prediction head. The model uses a hidden size of 96, intermediate size of 192, six transformer layers, six attention heads, and a maximum context length of 1024 tokens. Longer histories are truncated by prioritising more recent events.

### 3.3. Encoding Strategies

We evaluate five representative approaches for encoding continuous values in transformer-based EHR models: discretisation, combination, combined binning, concatenation, and FiLM (Feature-wise Linear Modulation) (Perez et al., 2018). These methods were selected as representative examples of the numeric encoding strategies most commonly explored in transformer-based architectures. As shown in Table 1, the methods differ along four design axes: (1) whether continuous values are preserved after preprocessing, (2) whether values are discretised through binning, (3) how numeric values are encoded, and (4) how numeric encodings interact with categorical token embeddings.

Across all approaches, numeric values are normalised using percentile-based min-max scaling, where the 1st and 99th percentiles define the lower and upper bounds. Discretised approaches map values to categorical tokens of the form VAL_$X$, where $X$ denotes the bin index. Continuous approaches project numeric values into $\mathbb{R}^d$ using a linear layer, following strategies similar to FT-Transformer (Gorishniy et al., 2021) and SAINT (Somepalli et al., 2022). Numeric representations are either inserted directly into the sequence as tokens or combined with categorical embeddings through concatenation or FiLM conditioning. Figure 2 provides an overview of the evaluated encoding schemes following the example in Figure 1.

#### 3.3.1. DISCRETISATION

In the **discrete** method, continuous values are discretised into bins and encoded as categorical embeddings inserted directly into the sequence. Pre-training follows a standard masked language modelling (MLM) objective using cross-entropy loss. Because values are represented as standard tokens, this approach requires no architectural modifications and remains naturally compatible with token-based transformer architectures, including GPT-style models. In addition, representing values as independent tokens natu-

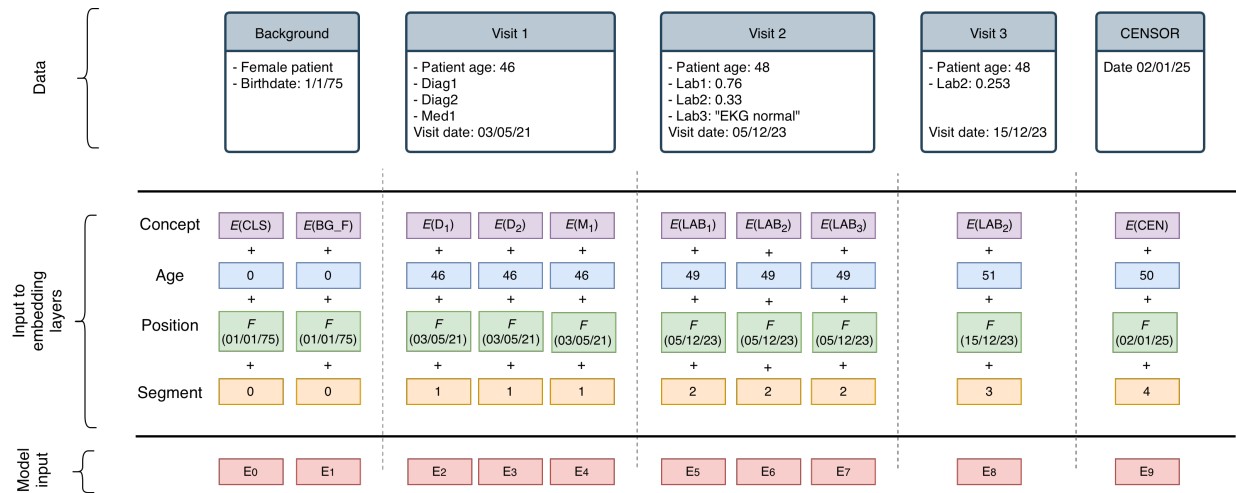

*Figure 1.* Illustration of a patient trajectory and the corresponding embedding layers forming the final model input. Here, $E(x)$ denotes the embedding function for concepts, and $F(x)$ denotes a function that maps raw timestamps to absolute positional encodings.

*Table 1.* Overview of the evaluated numeric encoding strategies.

| | Continuous values preserved | Binning | Value encoder | Sequence integration | Loss |
|---|---|---|---|---|---|
| Discrete | ✗ | ✓ | Categorical embedding | Token insertion | CE |
| Combination | ✓ | ✗ | Linear projection | Token insertion | CE+MSE |
| Combined binning | ✓ | ✓ | Linear projection | Token insertion | CE+MSE |
| Concatenation | ✓ | ✗ | Linear projection | Concatenation | CE+MSE |
| FiLM | ✓ | ✗ | Linear projection | FiLM | CE+MSE |

rally supports settings with multiple values per concept. However, discretisation reduces numeric precision and may discard fine-grained magnitude information.

### 3.3.2. COMBINATION AND COMBINED BINNING

In the **combination** approach, categorical and continuous features are embedded separately and inserted directly into the sequence. During pre-training, categorical and continuous values are jointly predicted using cross-entropy and mean squared error (MSE) losses, respectively. Both losses are combined with equal weighting. A sensitivity analysis of this weighting is provided in Appendix A.

Like the discrete approach, this method integrates numeric information directly into the token sequence, maintaining compatibility with token-based transformer architectures, including GPT-style models, although regression-based value prediction still requires some non-trivial adjustment. Unlike discretisation, values remain continuous, preserving finer-grained numeric information. Directly modelling floating-point values may nevertheless lead to less stable optimisation, particularly for heterogeneous or heavy-tailed value distributions (Sun et al., 2020; Pintea et al., 2023).

In the **combined binning** variant, values are first discretised into bins before projection through the linear embedding

layer. This hybrid approach aims to improve optimisation stability while preserving coarse-grained numeric structure, providing an intermediate trade-off between numeric precision and robustness.

### 3.3.3. CONCATENATION AND FiLM

In the **concatenation** and **FiLM** approaches, categorical and continuous representations are combined into joint embeddings. During pre-training, continuous values are masked whenever their associated categorical concepts are masked, and the model jointly predicts both components using cross-entropy and MSE losses.

In the **concatenation** method, categorical and continuous embeddings are concatenated and projected into a shared embedding space.

In the **FiLM** method, the joint embedding is computed using Feature-wise Linear Modulation (Perez et al., 2018):

$$\mathbf{e}_{\text{joint}} = \gamma(\mathbf{e}_{\text{cat}}) \cdot \mathbf{e}_{\text{cont}} + \beta(\mathbf{e}_{\text{cat}}),$$

where $\mathbf{e}_{\text{cat}}$ and $\mathbf{e}_{\text{cont}}$ denote categorical and continuous embeddings, respectively, and $\gamma, \beta$ are learnable functions conditioned on $\mathbf{e}_{\text{cat}}$.

These approaches preserve continuous information while explicitly conditioning values on their associated categorical

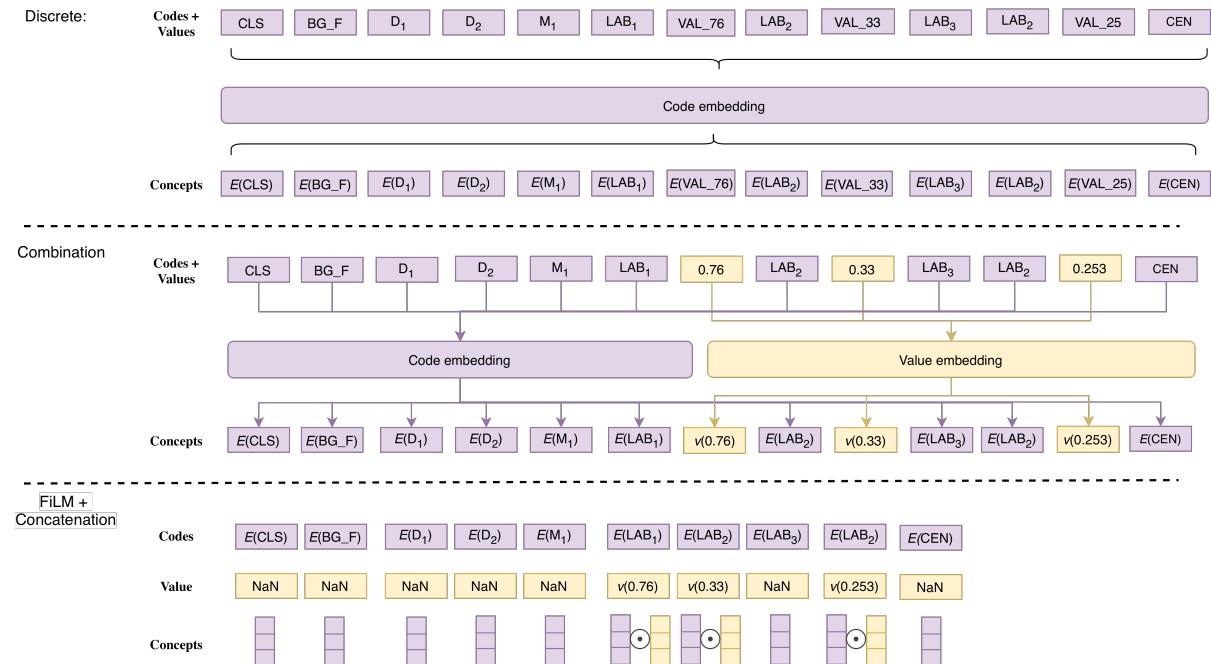

*Figure 2.* Numeric-integration strategies in EHR transformers following the example from Figure 1. Discrete: bin values into tokens. Combined: embed categorical and numeric features separately. Concat/FiLM: transform numeric values and merge them with categorical embeddings to create joint embeddings.

concepts. However, they introduce additional architectural complexity through explicit feature alignment and dedicated value representations, making extension to multiple values per concept less straightforward. In addition, sparsity from missing values may increase computational overhead as feature dimensionality grows.

### 3.4. Experimental Setup

We evaluate different strategies for incorporating numeric values using both synthetic and real-world data. To isolate the impact of value encodings, we insert synthetic lab measurements and corresponding outcome labels into each patient's timeline between birth and (if applicable) death, ensuring that numeric inputs appear in realistic temporal contexts. Each label is placed 10-180 days after the last synthetic lab measurement. Model inputs are then right censored before this label, such that the task is to predict the outcome using only the EHR history available prior to censoring.

All tasks used to evaluate the model are binary classification tasks. Because the synthetic laboratory values are the sole signal for predicting $y$, we compute the theoretical optimal AUROC for each setup, see Appendix B. Performance is then measured by a noise normalised version of the AUC, that we call information efficiency, $\text{info}_{\text{eff}}$, defined as

$$\text{Information Efficiency} = \max\left(\frac{\text{AUC}_{\text{model}} - 0.5}{\text{AUC}_{\text{opt}} - 0.5}, 0\right)$$

where $\text{AUC}_{\text{model}}$ denotes the AUC for a given model and $\text{AUC}_{\text{opt}}$ denotes the estimated theoretical AUC. This metric takes values $[0, 1]$ where 0 indicates random or worse performance, and 1 indicates that the model recovers all the signal theoretically available in the data. This therefore represents the fraction of recoverable information captured by the model, and can be used across difference datasets with varying noise levels.

#### 3.4.1. SYNTHETIC CLASSIFICATION EXPERIMENTS

The synthetic classification setup consists of a binary classification task under three class-conditional Gaussian settings with a fixed standard deviation of 10 and a varying mean separation. Each patient is assigned a binary label $y \in \{0, 1\}$ with equal probability. Conditioned on $y$, the patient is assigned a single synthetic laboratory value $x$ drawn from

$$p(x \mid y) = \mathcal{N}(\mu_0 + y(\mu_1 - \mu_0), 10^2),$$

where $(\mu_0, \mu_1) \in \{(35, 65), (45, 55), (48, 52)\}$. We fix the standard deviation at 10 (varying SD added no benefit over varying mean separation), and use a 50/50 class split. In this baseline, each patient has exactly one laboratory value ($n = 1$) and performance is evaluated as a function of the number of patients $N$. Experiments are conducted on datasets of size 10,000, 100,000, and 1,000,000. Each dataset is split into pre-training, fine-tuning, and test sets using a 50/40/10 split, respectively. All experiments are repeated five times. Based on performance in these

experiments, the best-performing and conceptually diverse methods are selected for subsequent arithmetic and clinical evaluations. These experiments additionally serve to characterise scaling behaviour and inform the dataset size used in subsequent arithmetic evaluations.

### 3.4.2. ARITHMETIC EXPERIMENTS

To assess the computational limits of each method, we designed a series of arithmetic tasks with increasing computational complexity, defined by how many values the model must attend to in order to complete the task. In all cases, patients are assigned binary labels according to whether a function $f(\cdot)$ of synthetic laboratory measurements exceeds its empirical median, ensuring a balanced binary classification problem. Formally, each task is defined as predicting

$$\mathbb{I}\left[f(\mathcal{L}) > \text{median}(f(\mathcal{L}))\right],$$

where $\mathcal{L}$ denotes the set of synthetic laboratory values present in the sequence. The specific function $f$ and structure of $\mathcal{L}$ vary by task:

- **Counting**. A single laboratory test type is repeatedly inserted into each patient sequence. The number of occurrences $m$ is drawn from a class-conditional Gaussian distribution, with $m \sim \mathcal{N}(\mu_0, 10^2)$ for positive patients and $m \sim \mathcal{N}(\mu_1, 10^2)$ for negative patients, rounded to a positive integer. Each occurrence is assigned an independent numeric value drawn from the same distribution for both classes. The task is to predict the label based on $m$. To prevent inference from sequence length alone, filler laboratory events are added so that all sequences have comparable lengths.

- **Addition**. Each sequence contains $n$ distinct laboratory tests, each appearing once with a numeric value. The task is to predict whether the sum $\sum_{i=1}^{n} \text{LAB}_i$ exceeds its median across patients.

- **Multiplication**. As in the addition task, each sequence contains $n$ distinct laboratory tests. The task is to predict whether the product $\prod_{i=1}^{n} \text{LAB}_i$ exceeds its median.

- **Polynomial evaluation**. Each sequence contains $n$ distinct laboratory tests. A polynomial score $s(\mathbf{x})$ of degree $d$ is computed over these values, and the label is determined by whether $s(\mathbf{x})$ exceeds its median. More details are available in Appendix C.

- **Trend and sharp-edge detection** Each sequence contains 3-10 laboratory values sampled from two Gaussian distributions with means $\mu_0$ and $\mu_1$. A patient is labelled positive if the sequence switches from one distribution to the other.

For the addition and multiplication tasks, we also introduce input noise to assess the robustness across methods. Performance for all arithmetic tasks is evaluated using info$_{\text{eff}}$, and each experiment is repeated five times. All experiments are conducted on datasets of 100,000 patients. This dataset size was selected following the synthetic classification experiments to provide a consistent evaluation setting across arithmetic tasks while remaining computationally feasible.

### 3.4.3. CLINICAL EXPERIMENTS

Finally, to assess whether synthetic findings generalise to real-world prediction, we evaluate the methods on breast cancer, lung cancer, and stroke, covering both long-term and acute outcomes. Models are pre-trained on data up to 2022 and evaluated on incident events in 2023, with prediction windows restricted to 3-12 months for cancer and 14 days - 12 months for stroke to avoid leakage. Experiments are conducted in patients aged 60+ (female only for breast cancer) to ensure sufficient incidence, and repeated five times. Dataset sizes vary across clinical tasks due to differences in incidence rates and cohort construction. The number of patients for each setup is provided in Appendix D.

## 4. Results

### 4.1. Deriving a General Binning Rule

To derive a generalised binning rule, we evaluate a range of bin counts $B \in \{3, 5, 10, 25, 50, 75, 100\}$ across varying dataset sizes and value distributions, using the synthetic classification setup described in Section 3.4.1. Unlike the main experimental evaluations, this analysis considers a wider range of dataset sizes $N \in \{10,000, 50,000, 100,000, 500,000, 1,000,000, 2,000,000\}$ to characterise scaling behaviour and derive a generalised power-law heuristic between dataset size and optimal bin count. In this setting, each patient contributes exactly one measurement for a given laboratory test, such that $N$ corresponds both to the total number of available observations and to the number of unique values for that lab. For each $(N, B)$ combination, experiments are repeated five times. For each $N$, the optimal bin count is selected as the geometric mean in log-space across repetitions, with uncertainty quantified by the SEM in log-space. Fitting a power-law relationship of the form $B(N) = aN^b$ yielded $B(N) = 1.14N^{0.237}$ with $R^2 = 0.83$ and exponent 95% CI: 0.086–0.387 (Figure 3). Logarithmic and linear models were also evaluated, but provided weaker fits to the observed relationship ($R^2 = 0.74$ and 0.53, respectively). However, given the limited number of evaluated dataset sizes and the wide confidence interval on the exponent, we present this relationship as an empirical approximation rather than a confirmed scaling law.

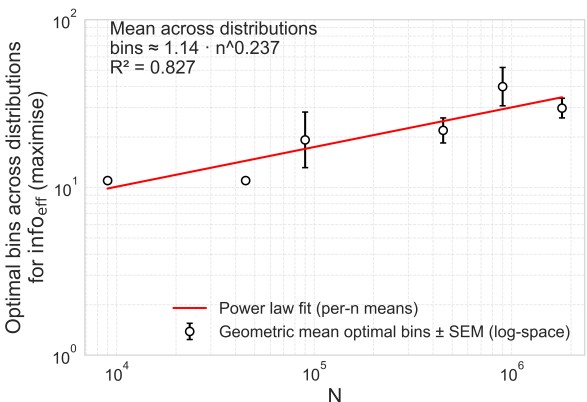

*Figure 3.* Power-law fit for the optimal bin count versus the number of unique values in the lab value distribution. Error bars indicate SEM.

### 4.2. Synthetic Classification Experiment

Figure 4 shows results for all five methods under the synthetic classification setup with mean separation $(\mu_0, \mu_1) = (48, 52)$. This setup highlights where the methods differ the most and is the most comparable to real clinical prediction scenarios as it reflects a relatively weak underlying signal. Results across all evaluated distributions are provided in Appendix E. From these results, we selected **FiLM**, **combined binning**, and **discrete** for subsequent evaluation as high-performing and conceptually distinct methods. Additional evaluation of the excluded methods show trends consistent with the primary analysis (Appendix I).

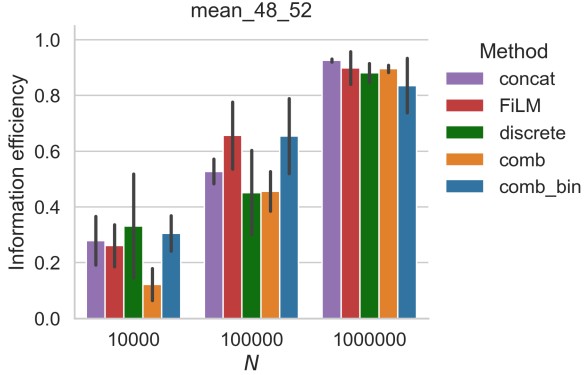

*Figure 4.* Synthetic classification performance ($d_\sigma$) for all five methods with mean separation $(\mu_0, \mu_1) = (48, 52)$. Error bars indicate SD.

### 4.3. Arithmetic Experiment

Figure 5 provides a high-level comparison of the methods across arithmetic tasks using rank-based aggregation. For each task, runs are grouped by the control variable (number of labs, noise level, or distribution) and ranked by $\text{info}_{\text{eff}}$,

with ties broken randomly. Ranks are then averaged first across repeated runs and then across all control-variable values to obtain a single mean rank per method. The mean rank is then rescaled as a percentage of the maximum observed rank and visualised in the spider plot. This summary highlights overall performance trends, but does not capture task-specific trade-offs. We therefore examine each arithmetic task individually.

Figure 6a reports performance on the counting (frequency) task across Gaussian class separations defined by $\mu_0$ and $\mu_1$. Figures 6b and 6c show performance on the multiplication task, where increasing task complexity eventually pushes methods to failure. Results for the addition tasks follow a similar pattern (Appendix F). Figure 6d shows model performance across polynomial degrees $d$. Results are averaged across runs with $n = 2, 3$, and 4 labs. Finally, Figure 6e shows performance on the sharp-edge task across multiple Gaussian separations.

Across the precision-sensitive arithmetic tasks, including addition, multiplication, and polynomial evaluation, FiLM produce the highest scores among the evaluated methods, particularly at lower noise levels and smaller input sizes. Statistical testing on the multiplication tasks show significant differences in most multiplication-with-noise settings and for multiplication tasks with $n_{\text{labs}} \leq 64$, although differences become less consistent at larger scales. To further investigate whether these effects stem from continuous value representations or explicit value-concept coupling, we additionally evaluate the combination method on a subset of arithmetic tasks (Appendix G). The combination method show intermediate behaviour between FiLM and discretisation-based approaches, suggesting that both continuous value representations and explicit value-concept interactions contribute to performance on precision-sensitive tasks.

*Figure 5.* Rank-based comparison of arithmetic-task performance using $\text{info}_{\text{eff}}$. Higher values indicate better overall performance.

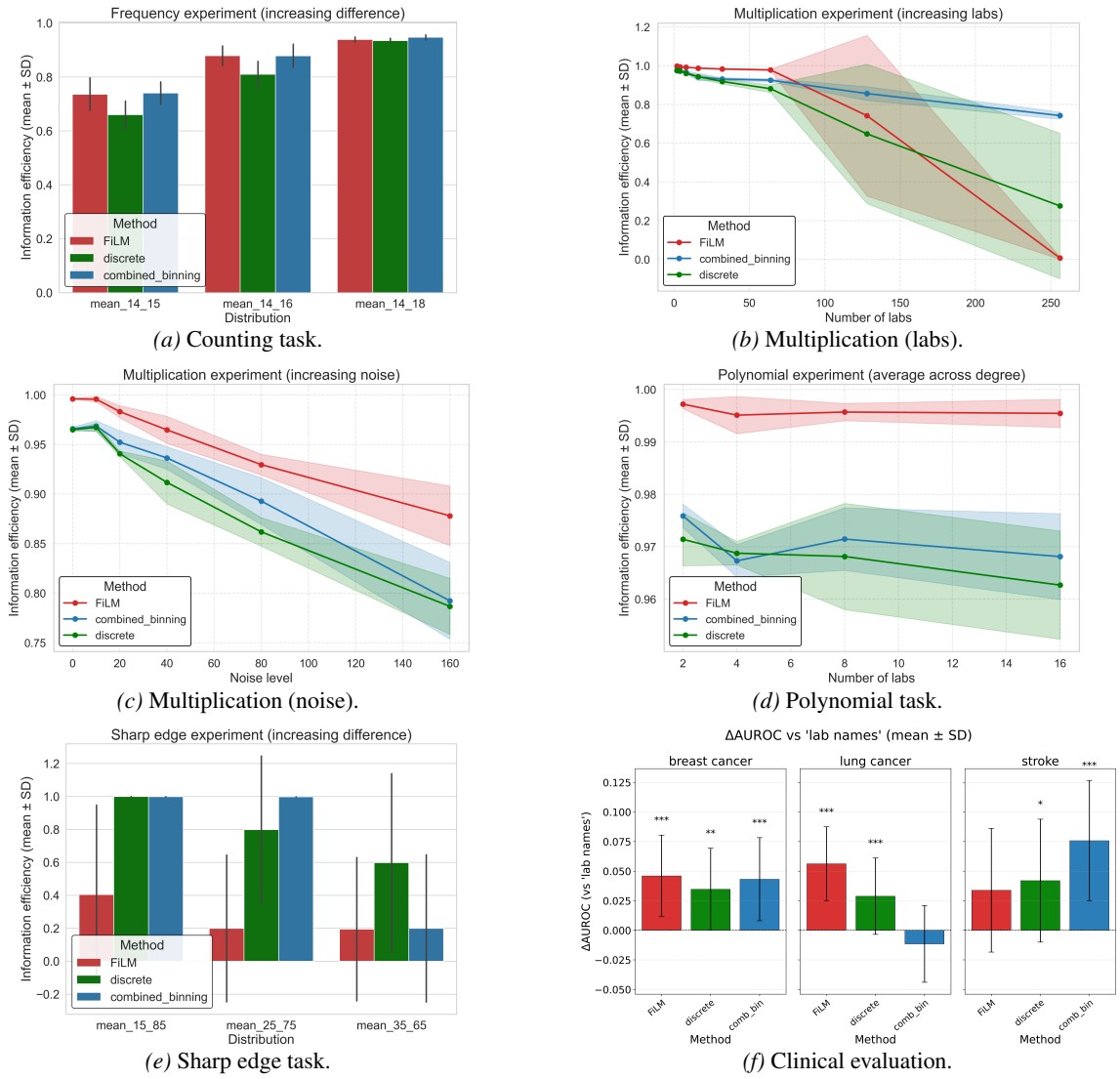

*Figure 6.* Performance across arithmetic tasks measured in info$_{\text{eff}}$ (Figures 6a–6e) and clinical prediction tasks (Figure 6f), showing AUROC change relative to the lab-name baseline. Error bars indicate standard deviation (SD) across runs. For the clinical tasks, stars indicate statistical significance relative to the lab-name baseline AUROC values (* $p < 0.05$, ** $p < 0.01$, *** $p < 0.001$; no star = not significant). Error bars indicate standard deviation (SD).

### 4.4. Clinical Experiments

Figure 6f shows the change in AUROC relative to a baseline model using laboratory names only. Across breast cancer, lung cancer, and stroke prediction, incorporating laboratory values yields modest and task-dependent effects, with no single method consistently dominating across tasks.

Because FiLM encodes each laboratory measurement using a single token, whereas the discrete and combined binning methods require two tokens per value, the methods operate under different effective context lengths within the fixed 1024-token budget. To assess whether this influence performance, we repeat the clinical experiments using a truncated FiLM setting with an effective context length matched to the

discretisation-based approaches. The truncated FiLM model shows performance similar to the original FiLM model, with no statistically significant difference ($p > 0.05$). This suggests that the observed differences are not primarily explained by access to longer patient histories, but instead reflect differences in the encoding strategies themselves. Results for the truncated FiLM model and the additional evaluated methods are provided in Appendix H.

## 5. Discussion and conclusion

Our results show that the choice of numeric value encoding scheme fundamentally shapes both performance and robustness in transformer-based EHR models, with different

methods occupying distinct points in the trade-off space between numeric precision, trend-based reasoning, optimisation stability, and architectural flexibility.

Across synthetic arithmetic tasks, FiLM, which retains continuous values and explicitly conditions them on categorical context, generally achieves the highest scores on precision-sensitive tasks. This is particularly evident in the multiplication and polynomial experiments, where preserving numeric magnitude and modelling value-concept interactions appear important. These findings suggest that, when sufficient data are available and architectural constraints permit, joint embedding approaches provide a useful inductive bias for value-sensitive reasoning. This result is notable in light of prior findings such as CEHR-BERT (Pang et al., 2021), where injecting additional tokens into the primary sequence outperformed auxiliary embedding layers. In contrast, our results suggest that for numeric values, explicitly modelling value-concept interactions through joint embeddings may provide advantages over sequence-level token integration specifically in precision-sensitive settings. A similar pattern was recently observed by Guo et al. (Guo et al., 2026), where joint event encodings outperformed factorised representations.

These gains in precision come with practical costs. Methods relying on aligned value layers introduce additional architectural complexity and scale poorly as the number of features or values per concept increases. In contrast, approaches that integrate numeric information directly into the token sequence without modifying the underlying architecture remain naturally compatible with encoder-only, decoder-only, and multimodal transformer models. Within this setting, applying binning before value projection provides an effective stabilisation mechanism in low-data regimes. We identify a simple discretisation heuristic whereby the empirically optimal number of bins approximately follows a power-law relationship with dataset size, offering practical guidance when domain-specific thresholds are unavailable. This stabilising effect is particularly relevant for EHR data, where numeric measurements are heterogeneous, sparsely observed, and often weakly predictive in isolation.

Despite prior reports that transformers struggle with numerical reasoning (Cho et al., 2025; Nogueira et al., 2021; McLeish et al., 2024), our models handle increasingly complex arithmetic tasks surprisingly well. In multiplication experiments, performance degrades smoothly rather than collapsing at the heuristic limit of $2^L$ with $L = 6$ being the number of transformer layers. This suggests that, for EHR-style tasks, approximate numeric reasoning may be sufficient, and that robustness and scalability are often more important than exact arithmetic precision. Clinical experiments reflect these trade-offs in a more constrained and noisy setting. Incorporating laboratory values yields modest

and task-dependent effects, with no single encoding consistently dominating across all outcomes. Differences between methods are further influenced by practical considerations such as sequence-length constraints, since approaches vary in how many tokens are required to encode each measurement.

Although the absolute AUROC improvements in the clinical experiments are modest, several methods produced statistically significant gains across specific prediction tasks. In practical clinical settings, even small improvements may be meaningful when applied at population scale, particularly for common outcomes and large screening cohorts. More broadly, our findings suggest that the practical value of numeric value integration lies not in a universally superior encoding strategy, but in understanding which encoding strategies provide stable behaviour under different modelling constraints and task regimes.

An additional contribution of this work is the introduction of a controlled evaluation framework for studying numeric value encodings approaches in transformer-based EHR models. By combining synthetic arithmetic tasks with real-world clinical prediction experiments, the proposed evaluation framework enables systematic comparison of encoding strategies across settings requiring different forms of numeric reasoning, robustness, and scalability. We make this evaluation framework publicly available to support future research on numeric encodings in clinical transformers.

This study has several limitations. First, the evaluated methods differ not only in how numeric values are encoded, but also in the surrounding architectural mechanisms used to integrate them, making it difficult to fully isolate the effect of the encoding strategy itself. Second, while the synthetic arithmetic tasks enable controlled evaluation of numeric reasoning behaviour, they do not fully capture the complexity of real-world clinical prediction. Consistent with this, performance differences across clinical tasks were generally modest and varied across settings. Nevertheless, the results suggest that multiple encoding strategies can provide sufficiently robust numeric reasoning for many EHR applications, with differences becoming most pronounced in settings requiring fine-grained numeric precision.

In summary, our findings suggest that when fine-grained numeric precision is required, joint encoding methods such as FiLM provide a strong overall trade-off between precision and robustness. In settings where architectural simplicity or scalability are primary concerns, hybrid token-based approaches such as combined binning offer a strong overall compromise, providing stable performance across tasks while retaining useful numeric structure.

## Acknowledgements

We thank the anonymous reviewers for their constructive feedback, which helped improve the manuscript. This work was supported by the Danish National Research Foundation, grant no. P1, and the Novo Nordisk Foundation, grant no. NNF23OC0083562.

## Impact Statement

Machine learning applied to electronic health records (EHRs) has the potential to make healthcare data more useful to clinicians, patients, and administrators. This study presents work on how numeric laboratory values can be encoded in transformer-based models for EHRs. Improved handling of numeric measurements addresses a key limitation in current EHR transformers and may support the development of more robust clinical prediction models, with possible benefits for early risk stratification and decision support. However, any downstream deployment would require careful validation across the intended patient population, as models trained on EHR data may inherit biases present in healthcare systems, and performance improvements on benchmark tasks do not necessarily translate into safe clinical deployment.

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

# A. Sensitivity analysis on weighted loss

As cross-entropy and MSE operate on different scales, equal weighting may bias optimisation. We therefore evaluated MSE weights $w \in [0.001, 0.01, 0.1, 0.5, 1, 1.5]$ for the combination method at $n = 100,000$, focusing primarily on down-weighting since the unbounded MSE may otherwise dominate optimisation. Across mean_35_65 and mean_45_55, performance remain stable for $w \in [0.1, 1.5]$, within one SD of the $w = 1$ setting. In mean_48_52, variability increased, but repeated runs show no consistent trend favouring any specific weighting. Overall, equal weighting appears robust.

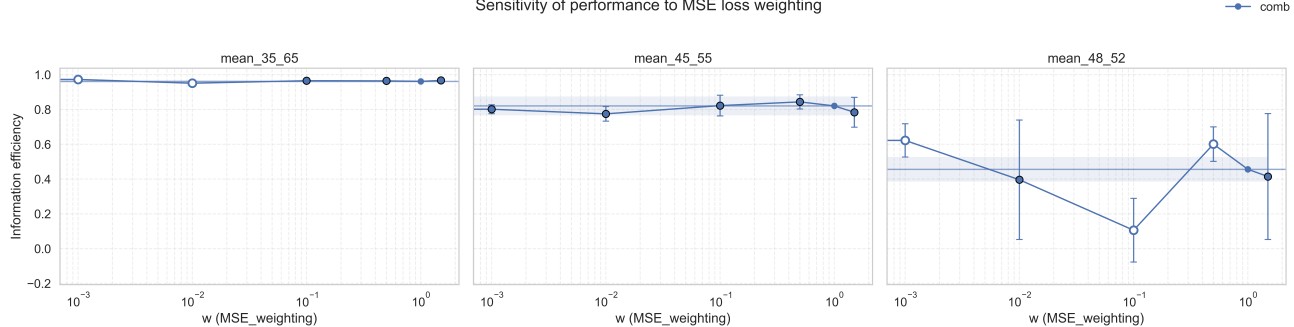

*Figure 7.* Sensitivity of performance to MSE loss weighting. The shaded band denotes the mean $\pm$ SD for $w = 1$, while error bars show SD for the remaining weightings. Black-edged markers indicate means within one SD of the $w = 1$ setting.

# B. Computation of theoretical AUC

Figure 8 shows the theoretical AUROC values for the baseline setting with $n = 100,000$. In this setting, the synthetic lab measurement is the only feature that carries information about the assigned label, which allows us to compute the theoretical maximum achievable AUROC. These theoretical values therefore represent the upper bound on performance for any model in this setup.

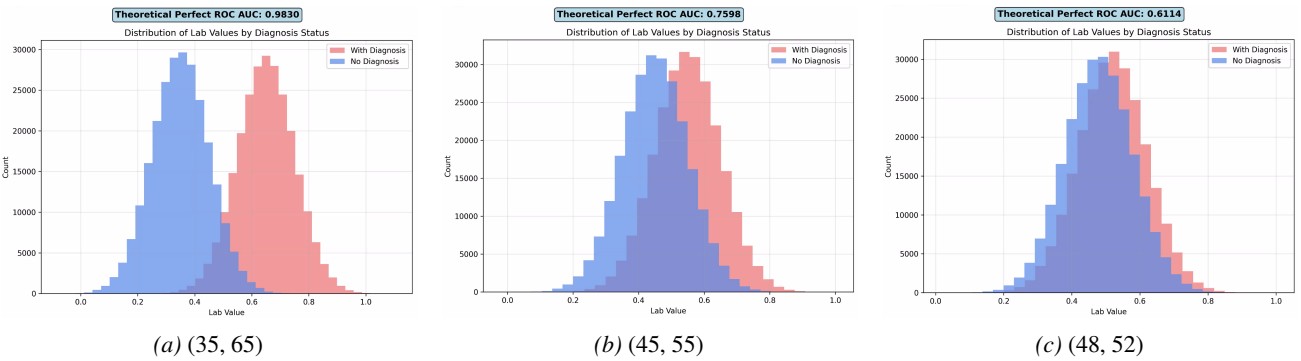

| (a) (35, 65) | (b) (45, 55) | (c) (48, 52) |

*Figure 8.* Theoretical AUROC for the baseline Gaussian experiments at different levels of class-mean separation.

# C. Polynomial task definition

For the polynomial evaluation task each sequence contains $n$ distinct laboratory tests each appearing once with values drawn from $\mathcal{N}(30, 10^2)$. A polynomial score is computed as

$$s(\mathbf{x}) = c_0 + \sum_{k=1}^{d} \sum_{\alpha \in \mathcal{A}(n,k)} c_\alpha \prod_{j=1}^{k} x_{\alpha_j},$$

where $d$ is the polynomial degree, $\mathcal{A}(n,k)$ is the set of non-decreasing index tuples $\alpha = (\alpha_1, \ldots, \alpha_k)$ with $\alpha_j \in \{1, \ldots, n\}$ (allowing repeated indices, e.g., $x_1^2 x_2$), and all coefficients $\{c_0, c_\alpha\}$ are sampled i.i.d. from $\mathrm{Unif}(-1, 1)$. The label is determined by whether $s(\mathbf{x})$ exceeds its median.

## D. Clinical Dataset Sizes

Table 2 summarises the number of patients used for pre-training, fine-tuning, and testing across the clinical prediction tasks. Differences in cohort size primarily reflect variation in disease incidence and eligibility criteria.

*Table 2.* Number of patients used in the clinical experiments.

| Task | Pre-training | Fine-tuning | Test |
|------|-------------|-------------|------|
| Breast Cancer | 1,093,329 | 116,176 | 28,884 |
| Lung Cancer | 1,093,329 | 226,292 | 56,684 |
| Stroke | 1,093,329 | 226,432 | 56,772 |

## E. Full Synthetic Classification Results

Performance across all synthetic classification distributions is shown in Figure 9.

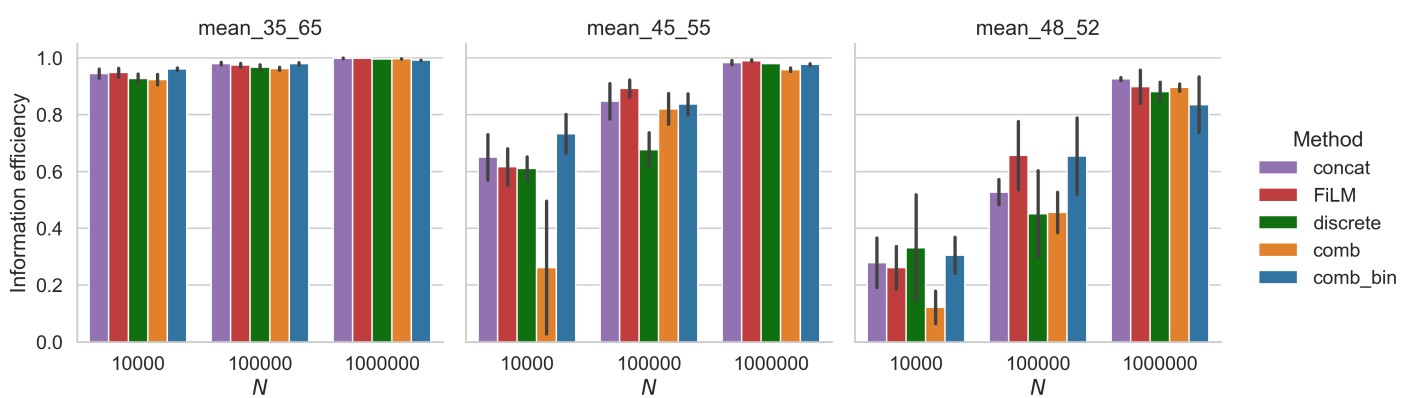

*Figure 9.* Synthetic classification performance ($d_\sigma$) for all five methods on all distributions.

## F. Results from addition tasks

Results from the addition experiments are shown in Figure 10a and Figure 10b. Figure 10a reports performance as the number of labs increases, while Figure 10b shows performance under increasing noise levels.

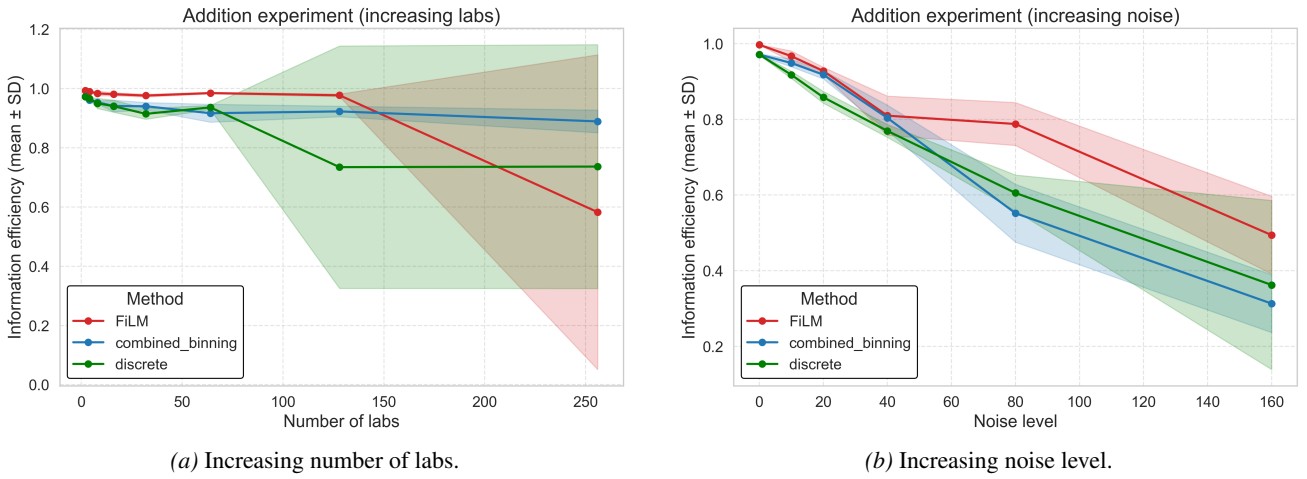

*(a)* Increasing number of labs.    *(b)* Increasing noise level.

*Figure 10.* Addition task performance across varying task difficulty.

# G. Arithmetic tasks with the combination method

To assess whether the performance gains of FiLM on arithmetic tasks come primarily from retaining continuous values or from jointly embedding values with their associated concepts, we evaluate a subset of arithmetic tasks using the combination method. As shown in Figure 11, the combination approach performs better than binning-based methods on precision-sensitive tasks, but consistently underperforms FiLM. This suggests that both continuous value encodings and joint value-concept modelling contribute to performance, with FiLM providing the strongest inductive bias for fine-grained numeric reasoning.

### Ranking Percentage Across Tasks

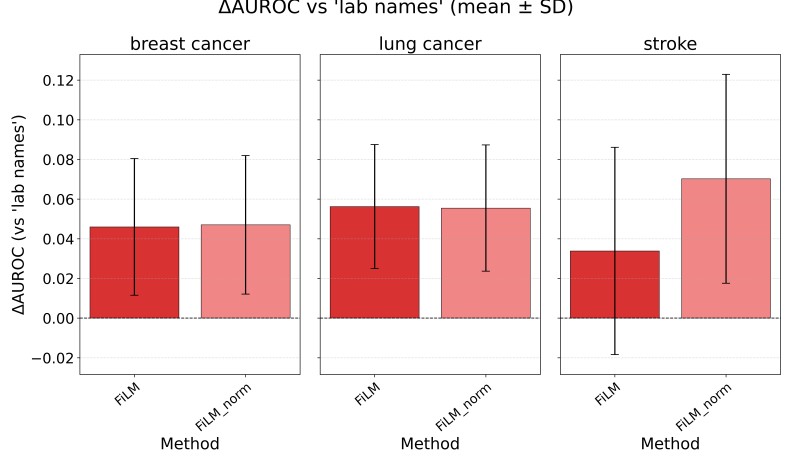

*Figure 11.* Spider plot of selected arithmetic tasks including the combination method.

# H. FiLM evaluation under matched token budgets

To evaluate whether FiLM performance was influenced by access to longer patient histories under the fixed 1024-token context window, we repeat the clinical experiments using a truncated FiLM setting with an effective token budget matched to the discretisation-based approaches.

*Figure 12.* Clinical prediction performance for the original and truncated FiLM models, shown as AUROC change relative to the lab-name baseline.

Figure 12 compares the original and truncated FiLM settings across the clinical prediction tasks. The truncated FiLM model

achieved performance comparable to the original FiLM model across all tasks, with no statistically significant differences ($p > 0.05$). These findings suggest that the observed FiLM performance is not primarily driven by increased effective context length.

To further assess the effect of truncation, Table 3 reports the percentage of truncated patients across clinical tasks and dataset splits. Figure 13 shows the distribution of patient sequence lengths for all patients prior to the 2022 clinical-task cutoff.

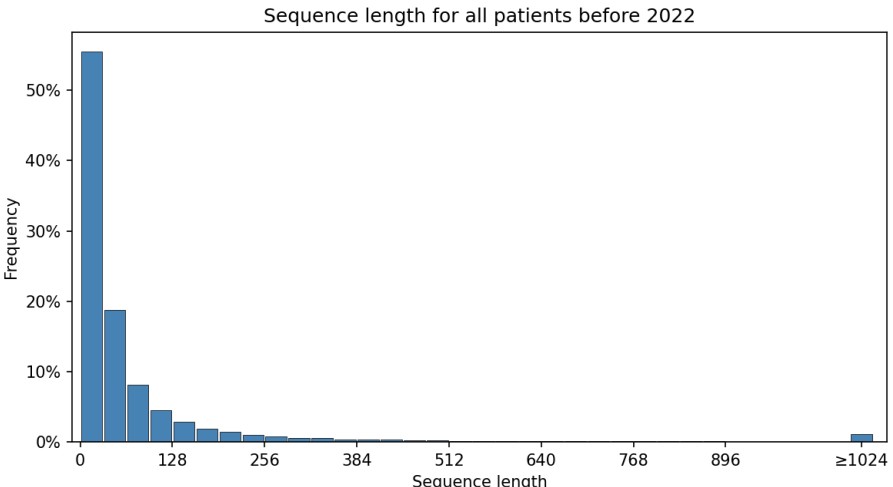

*Figure 13.* Distribution of patient sequence lengths prior to the 2022 clinical-task cutoff.

*Table 3.* Percentage of truncated patients across clinical tasks and dataset splits.

|  | Breast Cancer | | Lung Cancer | | Stroke | |
| --- | --- | --- | --- | --- | --- | --- |
|  | Fine-tune | Test | Fine-tune | Test | Fine-tune | Test |
| FiLM | 3.1% | 3.9% | 3.3% | 3.9% | 3.3% | 3.9% |
| FiLM-normed | 3.8% | 4.8% | 4.1% | 4.9% | 4.1% | 4.9% |

# I. Additional evaluation of excluded methods

To evaluate whether the exclusion of the combination and concatenation methods influence the overall conclusions, we additionally evaluated these methods on selected arithmetic and clinical tasks. The arithmetic subset includes multiplication as a representative precision-based task, and sharp-edge detection as a qualitatively different task involving distributional shifts over time.

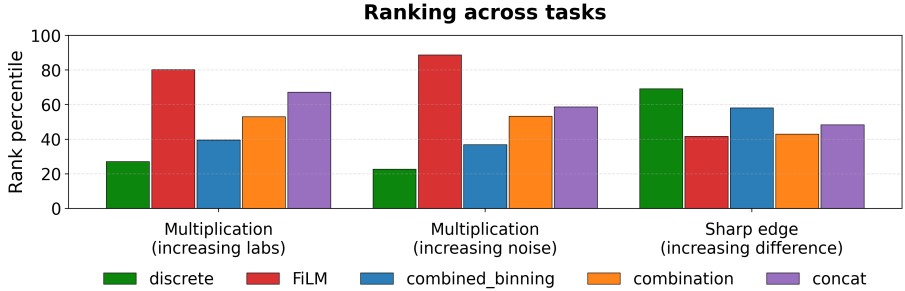

*Figure 14.* Relative ranking across selected arithmetic tasks for all evaluated methods.

Figure 14 summarises the relative ranking across the selected arithmetic tasks. Consistent with the primary analysis, FiLM performs best on the multiplication task, whereas discretisation-based approaches remain more robust on sharp-edge

detection. The combination and concatenation methods generally show intermediate behaviour and did not consistently outperform the primary methods selected for detailed evaluation.

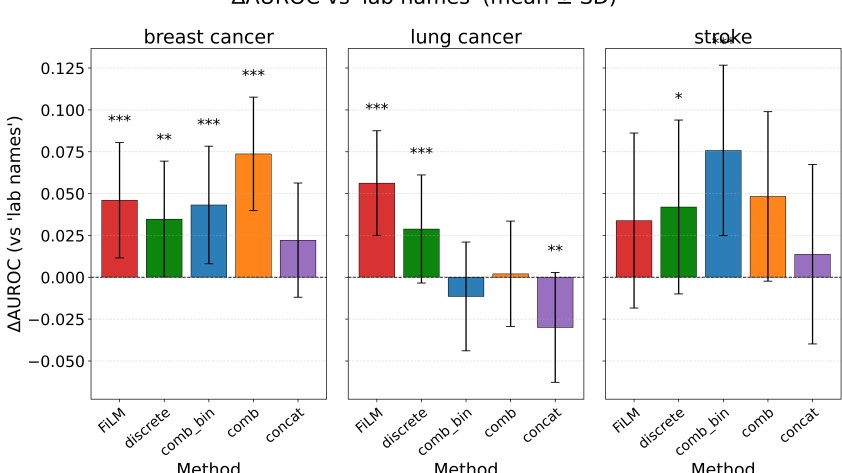

*Figure 15.* Clinical prediction performance for all evaluated methods, shown as AUROC change relative to the lab-name baseline.

Clinical results for all methods are shown in Figure 15. Across breast cancer, lung cancer, and stroke prediction, the additional methods produce modest and task-dependent effects similar to those observed in the primary analysis. In particular, the concatenation method shows a statistically significant decrease in performance for lung cancer prediction and does not provide consistent improvements across tasks. Overall, the additional evaluations do not substantially alter the conclusions of the main experiments, with relative performance trends remaining broadly consistent across settings.

## J. Software and Data

The codebase for the experiments, synthetic data generation, and evaluation framework is publicly available at: `https://github.com/Montgomeryyyy/BONSAI_values/tree/main`.

The real EHR data utilised in this study were acquired from hospitals in the Region of Zealand and the Capital Region of Denmark, which was approved by the Danish Patients Safety Board (Styrelsen for Patientssikkerhed, approval #31-1521-182) and the Danish Capital Region Data Safety Board (Videncenter for data-anmeldelser, approval #P-2020-180). Anyone wanting access to the data will be required to meet research credentialing requirements as outlined on the web site: `https://www.regionh.dk/til-fagfolk/Forskning-og-innovation/Hvilke-tilladelser-kraever-dit-projekt-/Sider/Forskningsprojekter-baseret-p%C3%A5-registerdata-og-journaldata.aspx`.

