# OpenReview forum: "How Should Transformers Encode Numeric Values in Electronic Health Records?"
_ICML.cc/2026/Conference — ICML 2026 regular_

### Official Review · Reviewer_peqc · 2026-02-22

**Soundness:** 2
**Presentation:** 3
**Significance:** 3
**Originality:** 3
**Overall Recommendation:** 3
**Confidence:** 4

**Summary:**

The authors observe that pre-trained EHR models often make limited use of numeric values, despite their importance. Although some existing approaches incorporate numeric information, this is typically done alongside many other architectural changes, making it difficult to isolate which representation strategies are actually effective. Common approaches include discretizing values into tokens or embedding them through linear layers, with the resulting representations combined via concatenation, summation, or by treating them as separate tokens.

To address this, the paper introduces a set of synthetic and real-world tasks designed to better expose differences between numeric representation methods. Across these evaluations, the authors suggest that FiLM is among the strongest approaches.

**Compliance With Llm Reviewing Policy:**

Affirmed.

**Final Justification:**

The authors addressed my concerns in the rebuttal and are committed to improving clarity of experiments and discussion of different embedding strategies.

**Key Questions For Authors:**

### A. Truncation (lines 145-146)
1. The model uses a context length of 1024. How does this compare to the distribution of sequence lengths in the data? In particular, how frequently does truncation occur? Including a histogram of sequence lengths in the appendix would help readers understand how much of the available clinical history is retained.
2. For the clinical prediction tasks, was truncation applied after truncation caused by censoring?

### B. Further questions
1. Were negative values of IE observed? The definition of the theoretical optimal AUROC (lines 210-219) uses 0 rather than allowing negative values. Is this choice motivated by prior work (if so, a citation would be helpful)? In principle, negative values could still correspond to a well-performing model after output negation and may therefore be informative.
1. Why was min-max normalization chosen? Given the presence of potential outliers in clinical data this normalization would be sensitive to, it could be a suboptimal choice.
1. Is combined binning (line 159) a previously proposed method? Unlike other techniques (e.g., FiLM), it does not appear to be cited, so clarification on whether this is novel or adopted from prior work would be useful.

**Limitations:**

There is little-to-none discussion of limitations in the paper, with authors mainly nothing that results would have to be re-verified before deployment. I would recommend also noting the inconclusivness of the results (as I already discussed in 'strenghts and weaknesses' - section D).

**Strengths And Weaknesses:**

## A. Strengths

**The overall approach taken by the authors is strong**, and has a potential to advance understanding of this area. In particular, I would highlight the following points:
- The paper addresses **an underexplored but practically important question**: how to quantify and compare different strategies for embedding numeric values in EHR models. Finding an answer to this question has **high potential impact**, and could be relevant for all future work on pre-trained EHR models.
- **A diverse range of methods is evaluated**, spanning multiple design axes, including discretization strategies, mechanisms for inserting numeric information into the sequence, and alternative embedding approaches for numeric values.
- The experimental setup uses controlled synthetic tasks which help isolate differences between methods and real-world prediction problems that show whether differences are significant in practice.
- **Experiments are repeated across multiple runs, enabling the reader to assess whether differences between models are statistically significant.**
- Code will be released, allowing for reproducibility of results.



## B. Weaknesses

### C. Presentation
While **the overall research direction is strong, the paper is noticeably weaker when it comes to presentation**. In particular, the authors could improve:

a) the discussion of existing representation methods (section 3.3),

b) the presentation and consistency of experimental results, and

c) figure quality and clarity of descriptions.

I outline specific concerns below.



Subsection 3.3 (Data Representations) is divided into several **subsubsections, but in practice these largely repeat similar ideas**. Specifically:

1. The loss functions are described multiple times (cross-entropy for discrete tokens and MSE for numeric values), making it unclear whether there are meaningful differences in how loss is computed across representations beyond the fact that MSE is not always applicable.
2. Many of the proposed representations appear to reuse a common set of components - discrete embeddings, linear projections of numeric values, quantization, concatenation or addition, and direct insertion of numeric tokens into the sequence. The role and significance of these components could be introduced once, after which individual methods could be described more concisely as combinations of these building blocks.

I believe **a clearer structure would be to first discuss the core components of embedding strategies and then describe each method** in a short paragraph **highlighting how these components are combined**. A summary table listing the evaluated methods (e.g., embedding strategy, quantization choice, and mechanism for integrating numeric values into the sequence) would substantially improve clarity. Similarly, **the description of loss functions could be consolidated into a dedicated section** that explicitly highlights any representation-specific differences.

Later in the paper, **the experimental setup varies considerably across sections with little explanation**. For example, the baseline experiments use dataset sizes of 10k, 100k, and 1M samples, the arithmetic experiments use 100k, and the clinical experiments do not clearly state dataset size. Section 4.1 further introduces six dataset sizes, which adds to this inconsistency. If comparisons across experiments are intended, **greater consistency would be helpful**.

Additional general issues:
1. In several places (including figures and main text), the notation VAL_X is used instead of properly formatting X as a subscript - $\text{VAL}_X$ (e.g., line 161, Figures 1 and 2).
2. **Error bars are shown in multiple plots without clearly specifying what they represent**. Both SEM and standard deviation are used for some figures, making it ambiguous for figures where it’s not explicitly stated. In general, figure descriptions are very brief  (e.g., Figures 4 and 6).
3. ‘Baseline experiment’ (section 4.2) doesn’t seem to be used as a baseline in following experiments. I would recommend referring to it as a ‘synthetic classification experiment’ instead.

### D. Soundness
In my opinion, **the conclusions drawn in the paper are stronger than what the evidence supports**, given the small differences observed between methods.

In the baseline experiments (Figure 4), a subset of methods is selected for further evaluation (lines 304–305). However, considering the magnitude of the reported errors, it is not clear that any method consistently outperforms the others with statistical significance (Note however that the meaning of error bars is not stated in the paper, meaning my conclusion could be wrong). I would encourage the authors to repeat at least a subset of subsequent experiments using the methods that were excluded at this stage. Since most arithmetic tasks yield similar performance across methods (with the exception of sharp edge detection), a reasonable compromise under compute constraints would be to evaluate all methods on the clinical task, the sharp edge detection task, and one additional arithmetic task.

More broadly, **the differences shown in Figures 4 and 6 appear small in absolute terms and may not be statistically meaningful** (again, the interpretation of error bars is unclear, but they appear relatively large). Despite this, the paper makes fairly strong claims about FiLM being the strongest approach (lines 349–354). The results instead seem largely inconclusive with respect to identifying meaningful differences between methods, particularly given the very similar performance observed on the real-world task - a point also acknowledged by the authors (lines 384 and 330).

#### E. Missing or incomplete results

1. Lines 301–302 claim that methods differ most when the means are close together, yet **the results for alternative mean configurations (referenced in line 262) are not presented**, even in the Appendix.

2. It would be useful to control for sequence length across methods to rule out differences caused by effective context length (as partially noted in lines 330-335). This could be done, for example, by inserting padding tokens after numeric values in FiLM-based approaches or truncating older events so that all methods operate with the same effective context length. This would help distinguish performance differences due to representation from those caused by context length.



## F. Conclusion

Overall, **due to the presentation and soundness issues in Sections 3, 4, and 5, the paper would benefit from substantial revision**. In particular, the authors should aim to ensure that:
- the presented results are sufficiently clear and convincing for readers to draw the same conclusions as the authors, and
- the experimental setup and discussion of results are easy to follow, including for readers who may not be deeply familiar with this specific line of work.

If presentation and soundness of the paper is sufficiently improved, I'll consider insreading my score.

---

> ### Author Rebuttal · Authors · 2026-03-30
>
> We thank the reviewer for the constructive feedback. All concerns will be addressed in the revision. Due to the character limit, the response below summarises the main revisions rather than addressing every point in full detail.
>
> We will restructure *section 3.3* by introducing common building blocks and describe each method as a combination of these components:
> | |Numeric values|Linear projection|Binning|Value as token|Interaction mechanism|Loss|
> |---|---|---|---|---|---|---|
> |Discrete|x|x|✓|✓|Attention|CE|
> |Combination|✓|✓|x|✓|Attention|CE+MSE|
> |Combined binning|✓|✓|✓|✓|Attention|CE+MSE|
> |Concatenation|✓|✓|x|x|Concatenation|CE+MSE|
> |FiLM|✓|✓|x|x|FiLM|CE+MSE|
>
>
>
> We will clarify the *experimental setup* and add a summary table. Variation in dataset sizes reflect different experimental objectives. Baseline experiments study scaling behaviour and inform the choice of 100k samples for arithmetic tasks as a balanced setting. Section 4.1 uses a wider range to analyse scaling, while clinical dataset sizes are determined by available data (~1.09M pretraining; ~116k-226k fine-tuning; ~29k-57k testing).
>
> The VAL_X *notation* is to stress that these are tokens, not continuous variables, and we will add to the figure captions that the *error bars* show standard deviations, except in Fig 3, where SEM is used.
>
> “Combination” and “concat” were initially excluded to reduce computational cost. We *extended the evaluation* to include all methods on the clinical tasks, the sharp edge detection task, and the multiplication-with-noise task. For clinical tasks (breast, lung, stroke), “combination” changes AUROC by  (0.069, 0.007, 0.056), significant for breast and stroke (p < 0.05). “Concat” changes AUROC by (0.018, -0.025, 0.021), significant for breast and lung (negative effect). Arithmetic results for combination are in the appendix; concat follows similar trends with slightly higher AUCs. These additions do not change the conclusions: differences remain modest, and relative trends are preserved. Notably, “concat” shows a statistically significant decrease in performance for lung cancer and does not provide consistent improvements across tasks. We will include these results in the revision.
>
> We agree that “FiLM shows strong performance” overstated consistency. Our intended claim is narrower: FiLM tends to perform better in precision-sensitive settings, not universally. To support this, we added significance testing on the multiplication tasks. In increasing-noise, FiLM outscores binning-based methods (discrete, comb_bin) across all noise levels with statistical significance (p < 0.05, often p < 0.01-0.001). In increasing-labs, FiLM performs better for $n_{labs}\leq64$, with significance from p < 0.1 to p < 0.001, while differences become less consistent at larger scales. Against combination, FiLM performs better in most noise regimes (4/6, p < 0.05) and in increasing labs for $n_{labs}\leq64$, though with weaker significance (p < 0.05-0.01).
> More broadly, our goal is not to identify a single best method, but to show that incorporating numeric values can be beneficial and that encoding choice matters in specific regimes. FiLM is favourable in precision-sensitive settings, but its advantages are less consistent and come with increased complexity. In contrast, combined binning provides stable performance across tasks while remaining simple and scalable. In practice, we do not claim a universally superior method, but suggest combined binning as a pragmatic default, reserving FiLM for settings requiring high numeric precision.
>
> The *figure illustrating the alternative mean configurations* was inadvertently omitted during manuscript preparation. We will include it in the revised manuscript.
>
> *Controlling for sequence length* is important to isolate encoding effects. We conducted additional experiments with matched sequence lengths across methods, and results remain consistent with our original findings (see response to Reviewer nCZk).
>
> Questions:
>
> (1+2). Truncation is applied once, after censoring, ensuring that only post-censoring sequences are subject to length constraints. It occurs relatively infrequently: depending on task, 3.5-3.6% of sequences for FiLM and concatenation, and 4.3-4.5% for the remaining methods. We will add a histogram of sequence lengths to the appendix for transparency.
>
> (4) The description in the manuscript was incorrect: we do not use strict min-max normalisation. Instead, we define bounds using the 1st and 99th percentiles rather than the absolute minimum and maximum, reducing sensitivity to outliers.
>
> (5) The combination method is inspired by SAINT and FT-Transformer, which use normalised continuous inputs. We introduce quantisation only as a stabilising preprocessing step. Values remain numeric (piecewise constant) and use the same linear embedding, so the downstream architecture is unchanged. Thus, “combined binning” is a preprocessing modification rather than a novel method.

---

> > ### Author Rebuttal · Reviewer_peqc · 2026-04-02
> >
> > Thank you for clarifying. Given the commitment of authors to improve the manuscript I'll increase my score.

---

> > > ### Author Response · Authors · 2026-04-03
> > >
> > > Thank you again for your thoughtful review. The character limit constrained our response somewhat, and we would be glad to address any open issues during the discussion phase.
> > > We remain happy to provide any further clarification on the experimental analyses or embedding strategy discussion if helpful.

---

### Official Review · Reviewer_4sHt · 2026-03-09

**Soundness:** 2
**Presentation:** 4
**Significance:** 2
**Originality:** 3
**Overall Recommendation:** 4
**Confidence:** 5

**Summary:**

The authors describe how the impoverished representation (sic) of numbers in EHR-specific transformer models constitutes a gap in making use of transformers, ultimately all the way into the clinic and lab IMSs. They continue to systematically investigate what constitutes a "good enough" approach, drawing the conclusion that the clinic is a long way off and that robustness and deployability requires more research.

**Compliance With Llm Reviewing Policy:**

Affirmed.

**Ethical Review Concerns:**

This is mostly to remind the authors that they must include all the specifics of the ethical permits necessary to obtain 2 million Danish EHRs, in full detail, upon publication of this paper anywhere.

**Ethical Review Flag:**

Flag this paper for an ethics review.

**Ethics Expertise Needed:**

["Responsible Research Practice (e.g., IRB, documentation, research ethics)", "Privacy and Security (e.g., personally identifiable information)"]

**Final Justification:**

Very happy with the reply to my last concern. I would like to increase my score of soundness from 2 to 3 (good).
Best of luck with final steps!

**Key Questions For Authors:**

I guess this data extraction is a result of "forskefabrikken" (I think it is called) and that your ethical permits were handled via that pipeline, as there is no mention of ethics in the paper? Just to make sure you include those permits, I will flag this as an ethical issue in my review.

How many synthetic measurements were added per patient? The 10–180 day gap between the last synthetic measurement and the label is quite a wide range. If the model sees variation in this gap, it could learn to use temporal distance as a feature rather than the numeric value itself. Is the gap randomised uniformly or held constant across conditions, and does the theoretical AUROC calculation account for any information leakage from the gap distribution?

**Limitations:**

The last contributions bullet in the intro covers an honest and important limitation, well captured.

I guess you observed the power law empirically, by optimising bin count across different dataset sizes and then fitting a curve? But empirical power laws are notoriously easy to claim and hard to confirm (see the work of Cosma Shalizi). You need to include, at minimum, the actual fitted exponent and its uncertainty, a comparison with at least log(n) and linear scaling as alternative hypotheses (a likelihood ratio test or similar), and a brief argument for why this scaling should generalise beyond the specific dataset and task set-up.

**Strengths And Weaknesses:**

Soundness:
Some biases are accounted for or at least recognised. Besides the subjective bias discussed, another source of bias is the one connected to sampling. When we draw blood from a patient, this is usually not done by a robotic nurse and how it is handled and cooled and stored varies immensely, sometimes strongly affecting the sample itself and how it is interpreted.

The aim is "practical default". Is a default not something that would emerge, or result from clinical guidelines or other operational constraints carefully thought through and negotiated amongst the practitioners very carefully, i.e. is it reasonable to have a default as a target?

CORE-BEHRT essentially sidesteps continuous numeric values, since it treats EHR data as sequences of discrete coded events and optimises the architectural and training choices around that representation. Lab values, vitals, and dosages are not in the model. They acknowledge this explicitly as a limitation, noting that these data sources are not easy to integrate. So, I am surprised you used this for inspiration.

Presentation:
The title is slightly problematic. Strictly speaking, transformers do not represent numeric values in EHRs. What they do is encode numeric values into high-dimensional embedding spaces, and the fidelity of that encoding depends heavily on the architecture and tokenisation strategy.

Obviously, the very first sentence is extra important and it set my mind off. Pausing there, I have been thinking a lot about how this map you refer to is not a map in any math sense, but rather an indirect and lossy relation. So when I read your first sentence I am thinking: "they can map it, but only if they bolt on machinery specifically designed for that purpose", and even then the transformer's own internal processing (attention, layer norms) does not natively respect numeric magnitude or ordering. The sequence-processing component treats everything as tokens in context, and any numeric understanding is emergent rather than architectural. Thus, I was curious to read on.

In the intro bit with successes and failures, are you referring to clinical validation or to methodological soundness and alignment with other processes and flows? In your intro, up to your contributions, you capture the status of the field very well, a strongpoint.

I am missing a 2.2 Laboratory Information Management Systems, explaining how learning approaches are now starting to be applied to LIMS, now that standardisation via OMOP, FIHR and other standards for lab flows are being rolled out.

The appendix is less of a traditional appendix, more a way of continuing the paper over the page limit.

Significance:
By the authors' own admission, not that high. But may well inspire more work.

Originality:
Good, even if there are things hard to follow and criteria for a particular choice is missing, the paper is easy to read and figures are very informative.

---

> ### Author Rebuttal · Authors · 2026-03-30
>
> We thank the reviewer for their thoughtful and detailed feedback, and appreciate the careful engagement with both the conceptual framing and practical implications of the work. Below we address the main points and outline revisions.
>
> **Soundness**
>
> *Sampling bias*
>
> We agree and will clarify this point in the manuscript by noting that laboratory values, while less subject to documentation bias, are still affected by measurement and handling variability.
>
> *Use of default*
>
> We use “practical default” to mean a modelling choice, not a clinical guideline. It serves as a robust baseline for integrating numeric data when task-specific guidance is unavailable, while still allowing tailored choices when domain knowledge exists. We will clarify this in the manuscript.
>
> *CORE-BEHRT*
>
> CORE-BEHRT was chosen as a strong, well-established baseline with public implementations, providing a reproducible starting point. Its lack of native numeric support aligns with our goal of systematically evaluating strategies for adding numeric values. Starting from such a baseline reduces the risk that gains reflect better tuning rather than the value encoding itself, and enables controlled comparison without additional design confounders. We update the architecture to a ModernBERT backbone to retain relevance while preserving the original design principles.
>
> **Presentation**
>
> *Title*
>
> Our intent was to describe how numeric information is incorporated into the model, not to imply that transformers inherently preserve magnitude or ordering. The goal of this work is to investigate encoding strategies that better align with the underlying numeric properties of the data, moving beyond purely discrete tokenisation. We will revise the title and introductory phrasing accordingly, e.g.: "How Should Transformers Encode Numeric Values in Electronic Health Records?" or "Encoding Numerical Values in Electronic Health Records for Transformer Models”.
>
>
> *Intro*
>
> We refer to both clinical validation and methodological comparability. Some approaches have not been evaluated in EHR settings, while others are tested under different setups, datasets, and assumptions. This makes it difficult to assess relative performance and the conditions under which methods succeed or fail. Our aim is therefore a more systematic, controlled evaluation across numeric data scenarios.
>
> *LIMS*
>
> This is a valid point. Our experiments use extracted data, but these originate directly from the production LIMS/EHR systems without manual curation or intermediate standardisation steps. This makes the setup directly relevant to how learning approaches are increasingly applied to operational laboratory data pipelines. We will add a brief note to clarify this.
>
> *Appendix*
>
> Given the study scope, some supplementary results were placed in the appendix, as is common at ICML/NeurIPS. We are happy to swap parts to the main text following suggestions by the reviewers.
>
> **Questions**
>
> *Ethical permits*
>
> We confirm that all necessary ethical approvals and data access permissions were obtained. Access to the data is strictly controlled and requires prior ethical clearance. These details were omitted due to the double-blind review process, as the permits are linked to specific institutions and researchers. We will include all relevant approvals in the final version upon author de-anonymisation.
>
> *Synthetic measurements*
>
> The number of added synthetic measurements varied by experiment. In the baseline, a single measurement was added. For most arithmetic tasks (counting, addition, multiplication, polynomial), the number increased with task difficulty, while the noise experiments used three measurements and the sharp-edge experiments used between 3-10 measurements. Importantly, class labels were assigned randomly (50/50), and the temporal generation process was identical for positive and negative patients. The gap between the last synthetic measurement and the label was sampled uniformly and independently of class, as was the timing of measurements in experiments with multiple values. Patients for whom the label would occur after death, were excluded. As a result, temporal distance does not carry predictive signal and cannot be exploited by the model.
>
> *Binning rule*
>
> We agree that establishing a true power-law relationship would require substantially more experimentation. Our intent was not to claim a universal scaling law, but to derive a practical heuristic summarising how the empirically optimal number of bins varies with dataset size. The power-law fit ($1.14 \cdot N^{0.237}$) achieved $R^2 = 0.83$ (exponent 95% CI: 0.086-0.387). Following the reviewer’s suggestion, we also evaluated alternative models: a logarithmic fit ($R^2 = 0.74$) and a linear fit ($R^2 = 0.53$). Given the wide confidence interval on the exponent, we avoid strong claims of a confirmed scaling law and instead present this as a simple, data-driven approximation for practical binning.

---

> > ### Author Rebuttal · Reviewer_4sHt · 2026-04-02
> >
> > I am glad to see my critique prompted many revisions, most of which I am sure will make the paper even more readable. Good that the ethical permits were omitted only for reasons of anonymity.
> >
> > Could the authors clarify "We update the architecture to a ModernBERT backbone to retain relevance while preserving the original design principles." and how this will be manifested in their submission? I am frankly unsure about if I should interpret the reply as an indicator of future work/upgrade, or if the authors are doing the change now already?
> >
> > The other revisions proposed look fine to me and if done well I would consider increasing soundness from 2 to 3.

---

> > > ### Author Response · Authors · 2026-04-02
> > >
> > > Thank you for your response and for highlighting this ambiguity. To clarify, the ModernBERT update is already part of the current paper, not a planned revision or future work. All experiments in the submission use the ModernBERT backbone while preserving the original design principles. The updated text will make this explicit.
> > >
> > > Does this answer your question? Please let us know if any part remains unclear.

---

### Official Review · Reviewer_nCZk · 2026-03-09

**Soundness:** 3
**Presentation:** 3
**Significance:** 3
**Originality:** 2
**Overall Recommendation:** 4
**Confidence:** 3

**Summary:**

The paper studies how numeric values should be represented in transformers, particularly in EHR data where lab measurements, vitals, and scores are continuous. It compares three broad embedding strategies, namely (i) discrete representations, (ii) continuous representations, (iii) hybrid approaches combining tokenization and numeric value information. Their evaluation strategy is twofold: (i) measure numeric reasoning ability, (ii) real-world clinical prediction. For the former, they use synthetic arithmetic tasks embedded in EHR sequences, and for the latter they use EHR data. Their findings include the following:

1) Architectures that explicitly model value–concept interactions perform best for tasks requiring numeric precision.

2) Hybrid token-based embeddings with binning offer a robust compromise between precision and training stability.

3) The optimal bin count scales with dataset size according to a power-law relationship.

4) Transformers generally perform approximate (“good enough”) numeric computation rather than exact arithmetic.

5) The benefit of including numeric lab values varies by clinical prediction task.

**Compliance With Llm Reviewing Policy:**

Affirmed.

**Final Justification:**

Following authors' rebuttal, I believe that the paper is now a well-executed and carefully validated empirical study. I am happy to increase my score to "weak accept" now. However, my main concern remains the limited methodological novelty, as the work primarily provides a systematic evaluation rather than introducing a new modeling approach.

**Key Questions For Authors:**

1) How do the different representation strategies handle missing numeric values, and could this affect the fairness of the comparisons?

2) Refer to Weakness 2. Are you sure that observed differences are due to numeric representation rather than architectural capacity differences? Is there a way to provide evidence for this?

3) Refer to Weakness 3. Why are cross entropy and MSE weighted equally? Did the authors evaluate sensitivity to this weighting?

4) Refer to Weakness 4. Can you repeat the experiments after fixing the number of tokens per measurement or expanding the context window?

5) Why use rank aggregation rather than directly comparing information efficiency scores across tasks? Are there any performance tradeoffs that the spider plot did not reveal?

6) The binning rule is derived from a small number of experimental points. How robust is the fit if (i) additional dataset sizes are included, (ii) other value distributions are used?

7) Given that clinical AUROC changes are modest, how should practitioners interpret the practical value of these methods?

**Limitations:**

Yes.

**Strengths And Weaknesses:**

Strengths
---

1) The manuscript addresses an important and underexplored issue in EHR transformers: how numeric values should be represented. Many existing models handle numeric data inconsistently or suboptimally, so a systematic investigation is valuable.

2) The synthetic + real hybrid experimental setup is stronger than typical EHR evaluations. The synthetic tasks allow controlled analysis of arithmetic ability, robustness to noise, and interaction complexity, while the real clinical tasks validate findings under realistic conditions.

3) The Danish EHR dataset covering approximately 2.2M individuals provides substantial scale and increases the credibility of the empirical evaluation (although individual experiments use different subsets).

4) The clear separation of representation strategies and the use of a noise-normalized metric (information efficiency) that adjusts AUC relative to the theoretical optimal AUC make comparisons across datasets and experimental setups more meaningful.

5) The empirically derived binning scaling rule relating optimal bin count to dataset size is interesting and arguably the most novel result of the paper.

Weaknesses
---

1) The main limitation is that the methodological contribution is primarily an evaluation framework and empirical analysis, rather than a new architecture or algorithm. For a conference such as ICML, this may limit the perceived novelty of the work.

2) The study claims to compare numeric representations, but the methods differ in architecture and training objectives. Therefore, it is difficult to attribute performance differences just to representation choice. The authors themselves say in Results section that “... both continuous values and explicit value-concept coupling contribute to performance.”

3) If I understood correctly, different methods use cross entropy for tokens and MSE for numeric regression, meaning the losses are fundamentally different. Equal weighting of these objectives may bias training.

4) As noted in the manuscript, FiLM encodes each laboratory measurement using a single token, while discretization-based approaches require two tokens per value. Under the fixed 1024-token context window, this effectively allows FiLM models to access longer patient histories before truncation occurs. Consequently, some performance differences in the clinical experiments may reflect differences in available context rather than representation quality. A more controlled comparison (e.g., equalizing token budgets or increasing context length) would help isolate the representation effects.

---

> ### Author Rebuttal · Authors · 2026-03-30
>
> We thank the reviewer for the detailed and constructive feedback and for recognising the value of our controlled synthetic-to-clinical evaluation and the practical importance of numeric value encodings in EHR transformers. Below, we address the main concerns and summarise the corresponding revisions.
>
> **Weaknesses**
>
> **W1** We agree that the contribution is not a new architecture, but a systematic evaluation of numeric value encodings. We address a gap in EHR modelling, where such design choices are rarely studied in a controlled manner, by providing unified comparisons and identifying regimes in which methods behave differently. The work includes a unified implementation, reproducible evaluation across synthetic and real-world data, and controlled synthetic generation to probe precision, robustness, and scaling.
>
> **W2** We agree that architectural and training differences make it difficult to attribute performance solely to numeric representation. Our goal was to compare methods as used in practice. Because representation, interaction, and input structure are tightly coupled, changing one typically requires changes to others, making strictly controlled comparisons difficult. To partially address this, we compared FiLM and the combination method on arithmetic tasks (Appendix). FiLM achieved higher ROC-AUC scores on precision-based tasks, suggesting contributions from both value representation and value-concept integration. We will clarify this limitation and retitle the paper around encodings.
>
> **W3** We agree that cross-entropy and MSE operate on different scales and that equal loss may bias optimisation. We therefore tested MSE weights $w$ (tested in [0, 0.001, 0.1, 0.5, 1.5]) on the combination method, focusing on downweighting since the unbounded MSE can otherwise dominate.  Across mean_35_65 and mean_45_55 performance was stable for $w \in [0.5, 1.5]$, within one SD of $w=1$. Outside this range, only slight increases in variability were observed. In mean_48_52, variability increased, but repeated runs showed no consistent trend favouring any specific $w$. Overall, equal weighting appears robust.
>
> **W4** We agree that tokenisation differences may affect effective context length. To address this, we repeated the clinical tasks with matched token budgets by truncating FiLM inputs to the same effective length as discretisation-based methods. Here FiLM showed similar performance to the original model, with no statistically significant difference (p > 0.05). This suggests that access to longer histories is unlikely to be the primary driver of the observed performance differences. We include these results in the revised manuscript.
>
> **Questions**
>
> **Q1** When a value is missing, only the laboratory test token is included without a numeric value. This is applied consistently across methods, preventing systematic differences between encodings due to missingness.
>
> **Q5** Direct comparisons of information efficiency across tasks were difficult to interpret because task difficulty differed. We therefore used rank aggregation to compare methods relatively within each task and summarise performance across tasks while reducing task-specific scale effects. We acknowledge that this does not capture effect magnitude, which is instead shown in Figure 6 alongside the ranking-based spider plot.
>
> **Q6** The rule is derived from a limited set of controlled synthetic experiments and is intended as a practical heuristic rather than a universal scaling law. While we observed a consistent trend across the tested settings, suggesting it may serve as a useful guideline in practice, we do not claim that this relationship generalises beyond them. Using this heuristic also avoids extensive optimisation over bin count, enabling a fairer comparison to methods without comparable hyperparameters. We agree that robustness to additional dataset sizes and value distributions is important, and broader evaluation would be needed to assess generality, particularly for real-world data. We will clarify this limitation in the manuscript.
>
> **Q7** To assess practical relevance, we increased the number of runs for the clinical experiments to enable statistical testing of AUROC differences. We found significant improvements for breast cancer across methods, and for lung cancer and stroke for selected methods.
> While the absolute AUROC gains are modest, improvements are observed across several tasks, although not uniformly across methods. In clinical settings, even small improvements can be meaningful at scale. For example, similarly sized improvements have been considered clinically relevant in established breast cancer risk models when adding breast density to Tyrer-Cuzick. The practical value of this work is therefore not a single large gain, but providing guidance on how to incorporate numeric values and identifying methods that offer stable performance across tasks, particularly in settings where laboratory values are expected to be informative.

---

> > ### Author Rebuttal · Reviewer_nCZk · 2026-04-04
> >
> > The authors have addressed most of my concerns satisfactorily.
> >
> > 1) W3: Addressed. The additional experiments demonstrate robustness to loss weighting. I am convinced this is not a significant issue.
> > 2) W4: Addressed. The controlled experiments with matched context length are convincing and remove this concern.
> > 3) W2: Partially addressed. I appreciate the clarification and additional analysis. While the coupling between representation and architecture is inherent in practice, this remains a limitation of the study, and the conclusions should be interpreted accordingly.
> > 4) Q1, Q5, Q6, Q7: Addressed. The responses are clear and satisfactory. In particular, the additional clinical analysis strengthens the case for practical relevance, and the clarification of the binning rule as a heuristic is appropriate.
> >
> > Final Score
> > ---
> >
> > Overall, the paper is now a well-executed and carefully validated empirical study. I am happy to increase my score to "weak accept" now. However, my main concern remains the limited methodological novelty, as the work primarily provides a systematic evaluation rather than introducing a new modeling approach.

---

### Official Review · Reviewer_vAqW · 2026-03-11

**Soundness:** 4
**Presentation:** 4
**Significance:** 2
**Originality:** 2
**Overall Recommendation:** 5
**Confidence:** 4

**Summary:**

This paper presents an empirical study on the effects of the method used for representing numerical values in transformed-based EHR models.

**Compliance With Llm Reviewing Policy:**

Affirmed.

**Final Justification:**

The authors responded to my concerns and clarified those.

**Key Questions For Authors:**

Does the study claim that the three ways picked for numerical repsrenation covers an exhaustive list of all possible ways for this? What about other choices?

**Limitations:**

Seems missing, in particular as referenced to real world data usage across multiple datasets

**Strengths And Weaknesses:**

Strengths
- The paper is well-organized, and the ideas are presented clearly.
- A list of rather substantive experiments is presented in the paper, and the derived conclusions are mostly supported by adequate evidence.

Weaknesses
- The study refers to “real” clinical data in a few places, but it’s unclear what exact real EHR data was used. Without actual data, the claims may be weakly supported.
- Relatedly, while the overall findings and reported patterns are rational and potentially informative, it is less clear to what degree the impact of these findings would be for readers and how significantly different the reported differences are.

---

> ### Author Rebuttal · Authors · 2026-03-30
>
> **Weaknesses**
>
> *Clarification on data*
>
> We apologise if this was unclear from the manuscript. The real-world EHR data used in this study are derived from the regional EHR system covering the Capital Region and Region Zealand in Denmark, and include all hospital interactions from 2016 to 2024 (2,218,028 unique individuals). This is described in the data section of the paper, but we will revise the text to make this more explicit.
> All data access was obtained through the appropriate approval processes, and further details will be included in the final version following author de-anonymisation.
>
>
> *Impact of findings*
>
> We thank the reviewer for this comment and agree that the practical implications should be stated more clearly.
> The primary goal of this work is not to establish a single universally superior method for numeric value representation, but to demonstrate that explicitly incorporating numeric values leads to measurable improvements compared to ignoring them, and to identify cases where certain encoding strategies are suboptimal.
> Within this scope, our findings provide two key insights: (i) incorporating numeric values is important for model performance, as demonstrated with the clinical experiments, and (ii) different encoding strategies exhibit trade-offs between precision, stability, and scalability.
> While we do not claim that any single method is universally optimal, we identify approaches that show favourable and consistent behaviour across settings. In particular, hybrid token-based methods such as combined binning provide a robust and scalable compromise, which we therefore prioritise in our software package and future clinical applications as a pragmatic choice rather than a claim of theoretical superiority.
>
> We will revise the manuscript to make these practical implications and design choices more explicit.
>
>
> **Questions**
>
> *Scope of methods*
>
> We do not claim that the three approaches considered in this study constitute an exhaustive set of all possible methods for numeric value encoding. Rather, our goal was to select representative approaches that capture the main design paradigms explored in the literature: discretisation-based methods, direct continuous embedding of numeric values via learned projections, and joint embedding or conditioning methods. These categories reflect common strategies used to incorporate numeric information into deep learning models and allow us to study the key trade-offs between precision, scalability, and architectural complexity in a controlled and interpretable way. This is also consistent with recent work that organises EHR tokenisation choices into broader design axes such as joint versus factorised encoding (Guo et al., 2026). We will clarify that our selection is representative rather than exhaustive and can add a summary table in the appendix describing the grouping of methods from papers.
>
> As this is an active area of research, additional methods may exist or have emerged since writing. If the reviewer is aware of specific approaches that would meaningfully extend this comparison, we would be happy to consider including them or discussing them in a revised version.

---

> > ### Author Rebuttal · Reviewer_vAqW · 2026-04-04
> >
> > Moved up the score. Thanks!

---

### Review · Ethics_Reviewer_43Xc · 2026-03-31

**Recommendation:** Remediation action needed

**Ethics Issue:**

Per reviewer 4sHt's ethical concern, the release or report of potential demographic information in real-world EHR data will improve the transparency of data usage in LLMs' evaluation.

**Remediation Action:**

Recommend adding a few sentences to discuss potential demographic information in the real-world EHR data, in the limitations or ethical considerations.

---

### Decision · Program_Chairs · 2026-04-30

**Decision:**

Accept (regular)

**Comment:**

## Paper summary

This paper examines the problem of how to represent numerical values from electronic health records (EHR) within transformers. The paper contributes a systematic comparison of discrete, continuous, and hybrid approaches. The claimed findings include:

* a characterization of the precision-stability trade-off
* evidence that "good enough" approximate numeric reasoning is often sufficient
* finding that gains from incorporating numeric values are modest and task-dependent.

Experiments first look at synthetic signals, assessing whether transformer representations can approximate a true decision rule in which a fake patient's sum/product/trend of lab values over visits exceeds the population median. Next, they examine real EHR data from over 2 million patients from the Capital region of Denmark. The task here is to predict onset of cancer or stroke, given a window of EHR visit data from 1 year to 3 months prior.

## Reviews and Discussion summary

The paper received 4 reviews that offered a range of initial assessments.

On the positive side, the paper:

* took on a problem that is "important and unexplored" (nCZk) and "underexplored" (peqc)
* used an approach that is overall "strong" (peqc)
* experiments that were "rather substantive" (vAqW) and "stronger than typical" (nCZk), with extra strength due to being "repeated across multiple runs" (peqc)

Weaknesses raised initially include:

* peqc: concerns about presentation (detailed suggestions to make more unified, less redundant)
* peqc: concerns about experimental conclusions stronger than the evidence supports
* vAqW: questions about how significantly different the reported differences are
* 4sHt: power laws in the binning rule need more justification
* nCZk: concern about lack of methodological novelty, though the paper is focused on evaluation
* nCZk: many questions about experimental details

The authors provided detailed responses to each reviewer. During the response period, all reviewers indicated main concerns were "fully resolved" (vAqW, peqc, nCZk) or "partially resolved" (4sHt), with all voices reaching consensus for weak acceptance and only relatively minor issues remaining. The partial aspect of 4sHt refers to a question about ModernBERT that does appear to my eyes resolved in later comments by the authors.

## Decision

I am happy to back the four reviewers in voting to accept: this work to my eyes meets all four criteria (soundness, originality, significance, and presentation clarity). I appreciate the careful investigation here and the use of synthetic data to tease apart specific aspects of numeric represenation.

I strongly encourage the authors to fulfill all promised revisions, and to fulfill their promise to provide sufficient documentation of ethical approvals needed to use their real EHR data (which were understandably omitted for anon review).